# Derivation of Novel Imaging Biomarkers of Neonatal Brain Injury Using Bedside Diffuse Optical Tomography: Protocol for a Prospective Feasibility Study

**DOI:** 10.3390/neurosci6030060

**Published:** 2025-06-30

**Authors:** Sabrina Mastroianni, Anagha Vinod, Naiqi G. Xiao, Heather Johnson, Lehana Thabane, Qiyin Fang, Ipsita Goswami

**Affiliations:** 1Michael G. DeGroote School of Medicine, McMaster University, Hamilton, ON L8S 4L8, Canada; sabrina.mastroianni@medportal.ca; 2Department of Psychology, Neuroscience & Behaviour, McMaster University, Hamilton, ON L8S 4L8, Canada; vinoda@mcmaster.ca (A.V.); nx@mcmaster.ca (N.G.X.); 3Division of Neonatology, Department of Pediatrics, Faculty of Health Sciences, McMaster University, Hamilton, ON L8S 4L8, Canada; johnsh30@mcmaster.ca; 4Department of Health Research Methods, Evidence and Impact, McMaster University, Hamilton, ON L8S 4L8, Canada; thabanl@mcmaster.ca; 5Department of Engineering Physics, McMaster University, Hamilton, ON L8S 4L8, Canada; qiyin.fang@mcmaster.ca

**Keywords:** functional near-infrared spectroscopy, diffuse optical tomography, hypoxic–ischemic encephalopathy, neonatal brain injury, feasibility studies

## Abstract

Prognostication of neurodevelopmental outcomes for neonates with hypoxic–ischemic encephalopathy (HIE) is primarily reliant on structural assessment using conventional brain magnetic resonance imaging in the clinical setting. Diffuse optical tomography (DOT) can provide complementary information on brain function at the bedside, further enhancing prognostic accuracy. The predictive accuracy and generalizability of DOT-based neuroimaging markers are unknown. This study aims to test the feasibility of prospectively recruiting and retaining neonates for 12 months in a larger study that investigates the prognostic utility of DOT-based biomarkers of HIE. The study will recruit 25 neonates with HIE over one year and follow them beyond NICU discharge at 6 and 12 months of age. Study subjects will undergo resting-state DOT measurement within 7 days of life for a 30–45-min period without sedation. A customized neonatal cap with 10 sources and eight detectors per side will be used to quantify cortical functional connectivity and to generate brain networks using MATLAB-based software (version 24.2). The Ages and Stages Questionnaires—3rd edition will be used for standardized developmental assessments at follow-up. This feasibility study will help refine the design and sample-size calculation for an adequately powered larger study that determines the clinical utility of DOT-based neuroimaging in perinatal brain injury.

## 1. Introduction

Hypoxic–ischemic brain injury during the birth process is manifested as neonatal encephalopathy after birth, termed hypoxic–ischemic encephalopathy (HIE) [1]. Estimated to occur in 1–7/1000 live births [2], it is a significant cause of neonatal death and cerebral palsy, epilepsy, hearing loss, and visual impairment [3,4]. Induced therapeutic hypothermia initiated within six hours of birth substantially reduces risk of death and moderate–severe neurodevelopmental disability at 18 months, with a number needed to treat of 7 [5]. Unlike motor impairment, hypothermia does not reduce the risk of neurocognitive deficits that may not be evident until school age [6]. Even in the absence of cerebral palsy, neonates with HIE exposed to hypothermia therapy are at risk of having an IQ score of <70 (11%) or 71–84 (32%) at school age [7]. Although not as debilitating as cerebral palsy, abnormalities of cognitive functions, such as attention deficits, learning disabilities, and abnormalities of visuospatial processing and executive functioning, will have profound and long-lasting negative impacts on a child’s language skills, academic performance, and quality of life. Being able to anticipate the risk of neurologic deficits in early infancy, the crucial formative period of brain development, may facilitate early interventions and improve outcomes. Early developmental intervention programs may include targeted speech–language therapy, physical therapy, parental coaching, parent–infant relationship enhancement, infant stimulation, psychological therapy, sensory modulation, and biofeedback therapy [8,9,10,11].

Neurocognitive functions are challenging to prognosticate at early stages of life, and predictions based on conventional magnetic resonance imaging (MRI) brain images are often erroneous [12]. The neural correlate of higher-order cognitive functioning in early development remains largely unknown. Previous research has described the role of functional connectivity between cortical brain regions as an important determinant. For example, the prefrontal and parietal cortex neural networks may support attentional control abilities [13]. Altered structural and functional connectivity between key brain regions results in impairment of related cognitive functions [14]. Quantifying the functional connectome can shed light on brain reconfiguration following a birth injury during the first 2 years of life [14]. Traditionally, morphologic assessment of conventional brain MRI (T1/T2-weighted) and diffusion-weighted imaging (DWI) performed at 3–5 days of life has been used in clinical settings to aid clinical decisions and parental counselling [15]. However, caution is warranted when using conventional MRI to predict future cognitive outcomes. Recently, it was reported that among infants with no or minor injury on MRI, 34% develop moderate delay, and 5% develop severe delay at 24 months [16]. Furthermore, neurophysiological studies conducted during the neonatal period predicted later psychomotor impairment in infants with normal MRI results [12]. Currently, physicians primarily depend upon observing the infant’s behavior over time to prognosticate motor and cognitive outcomes, which are biased by practice setting and experience. Such perceptions are subjective and vulnerable to errors. Since physicians’ estimates of neurodevelopmental prognosis can influence parental counselling and early interventions, validated tools are needed to provide more objective estimates.

Diffuse optical tomography (DOT) is a non-invasive optical neuroimaging method that can be applied at the bedside to assess functional connections between brain regions [17]. It is an extension of fNIRS (functional near-infrared spectroscopy) that combines hemodynamic information from dense optical sensor arrays over a wide field of view. This technique utilizes near-infrared light to detect synchronous and spontaneous fluctuations in cerebral blood flow across different brain regions [18]. It combines cap-based multichannel data acquisition, such as electroencephalography (EEG), with spatial resolution high enough to create detailed cortical brain maps [19]. It helps quantify region-specific brain activity at rest, like functional MRI (fMRI). Advantages of DOT over fMRI include that it is portable, is lightweight, has high tolerance to body motion, does not produce noise, and does not require patient sedation. Moreover, fMRI is not routinely available in clinical practice, where conventional MRI images are commonly used for prognostication. The latter provides a qualitative assessment of brain injury rather than a functional assessment. Although DOT images are inferior to those acquired using fMRI in terms of spatial resolution, DOT provides biochemically specific, time-resolved, and spatially localized information on cerebral processes [20,21]. It is a reliable tool for evaluating cerebral oxygenation, with an intraclass coefficient of 0.7–0.8 [22,23,24,25,26]. The reproducibility of assessing brain networks was previously demonstrated [27]. The tool’s safety has been assessed in non-sedated term and preterm neonates at the bedside [28,29,30,31,32]. Therefore, DOT can be potentially used at the bedside as an adjunct to other currently available clinical prognostic markers.

DOT-based functional neuroimaging may provide early recognition of neural deficits at the bedside after several perinatal events such as premature birth, intraventricular hemorrhage, and birth trauma. This tool can be used to analyze the effects of bedside interventions and individualized care plans, such as music therapy, reading programs, sensory modulation techniques, and sleep, on the functional maturation of the developing brain. This study aims to conduct a single-center pilot project to assess the feasibility of a larger-scale study investigating the prognostic value of neonatal brain functional connectivity measures using DOT to predict long-term neurological outcomes. The primary objectives of this pilot study are to (i) demonstrate the feasibility of recruitment and retention strategies along with data collection methods in the NICU and (ii) assess the fidelity and safety of DOT measurements in neonates. The secondary objectives of this pilot study include (i) identifying logistic challenges, unexpected adverse effects, or technical difficulties, (ii) gaining stakeholder engagement, including parents and NICU nurses, and (iii) providing preliminary data on the variability of outcome measures to calculate the sample size of the larger trial accurately.

## 2. Materials and Methods

### 2.1. Study Design

This is a single-center, prospective observational cohort study. The study design conforms to the modified Standard Protocol Items: Recommendations for Interventional Trials (SPIRIT) [33].

### 2.2. Study Setting

This study will be conducted at the McMaster Children’s Hospital neonatal intensive care unit (NICU). The province of Ontario, Canada’s perinatal services are organized into three levels, namely the following: (i) Level 1: low neonatal risk, postnatal care of healthy newborn and resuscitation/stabilization of ill infants before transfer; (ii) Level 2: neonatal care for gestational age > 32 weeks and birth weight > 1500 g [exception: Level 2c > 30 weeks and >1200 g]; and (iii) Level 3: neonatal intensive care for any gestational age and birth weight, availability of subspecialty consultants, and surgical capability. Since therapeutic hypothermia resources are available solely in Level 3 hospitals in Ontario, Canada, neonates born across Level 1 and 2 hospitals are transferred to Level 3 with a specialized neonatal transport team [www.pcmch.on.ca, 2 June 2025]. McMaster Children’s Hospital is a Level 3 unit attached to a high-risk maternal and fetal medicine unit. Approximately 50–60 neonates per year are admitted to the NICU with the diagnosis of HIE; ~90% of them receive therapeutic hypothermia, and ~80% of them are outborn neonates. Outborn neonates may be born in any of the surrounding Level 1 or Level 2 hospitals in the catchment area of McMaster Children’s Hospital. When indicated, passive cooling is started in the referring hospital, and the neonate is transferred immediately to the Level 3 unit to receive hypothermia therapy using a servo-controlled device. A two-member in-house neonatal transport team facilitates the transfer of neonates from the referring center to the Level 3 NICU.

### 2.3. Therapeutic Hypothermia Protocol

Therapeutic hypothermia is the standard of care for moderate to severe HIE. The severity of encephalopathy is graded using a modified Sarnat Staging [34] based on 6 domains: level of consciousness, spontaneous activity, posture, tone, primitive reflexes (suck and Moros’ reflex), and autonomic nervous system (heart rate, pupils, and respiratory effort). At McMaster Children’s Hospital, a servo-controlled device, CritiCool^®^, Belmont Medical Technologies Ltd., Billerica, MA, USA, is used for neonatal hypothermia therapy. The infant is wrapped in a temperature-regulated cooling blanket to reduce core body temperature to 33–34 °C and maintained for 72 h using a rectal temperature probe for continuous monitoring. Then, the neonates are rewarmed to 36.5–37.5 °C over 6 h. Electrolyte levels, fluid balance, bleeding tendency, and cardiorespiratory functions are closely monitored during the therapy. Additionally, brain function is routinely monitored with continuous amplitude-integrated EEG or continuous video EEG until rewarming is completed to detect and manage seizures. All neonates will undergo structural brain MRI and MR spectroscopy on a 1.5 Tesla MRI scanner (Siemens Healthcare Limited, Forchheim Germany) at a median age of 3–5 days. MRIs conducted in the clinical setting include sequences T1W, T2W, DWI, SWI, and, if indicated, MRV/MRA. After discharge, all infants are followed in the Developmental Follow-Up clinic for 2–3 years to detect and treat developmental delays.

### 2.4. Research Ethics Approval

This study has been reviewed and approved by the Hamilton Integrated Research Ethics Board (HIREB #13982). The study was also registered on www.clinicaltrials.gov on 18 August 2022, and last updated on 3 October 2024 (NCT05514665). Any changes in protocol modifications will be communicated to the HIREB by the principal investigators.

### 2.5. Consent to Participate

All substitute decision makers for neonatal participants will provide written informed consent, obtained by a clinical research coordinator. Any modifications to the protocol that may impact the study’s intervention, target enrollment group, and outcomes will result in a formal amendment to the protocol.

### 2.6. Eligibility Criteria

Newborns admitted to the NICU will be enrolled if there is (i) evidence of an acute perinatal event, fetal metabolic acidosis, and a neurological exam indicative of encephalopathy, (ii) birth weight ≥ 1800 g, and (iii) gestational age at birth ≥ 35 weeks. The enrollment plan and exclusion criteria are illustrated in Figure 1.

### 2.7. Interventions

#### 2.7.1. Study Equipment

In this study, we will use a multichannel fNIRS system—the Brite MKII (Artinis Medical Systems, Einsteinweg, The Netherlands)—a wearable and flexible device designed to measure cerebral oxygenation. Each unit can record up to 27 channels, with sensors (comprising both transmitter and receiver elements) arranged on a recording cap. Two devices will be used concurrently to achieve comprehensive coverage of the neonatal scalp. By positioning channels across distinct brain regions, we can simultaneously monitor neural activity in these areas. Each transmitter optode incorporates light-emitting diodes (LEDs) that emit near-infrared light at wavelengths of 760 nm and 850 nm. These wavelengths are used to estimate changes in oxygenated and deoxygenated hemoglobin. The system operates in continuous-wave mode at a frame rate of 10 Hz. The imaging cap is engineered with preconfigured apertures for sensor placement, which connect to a compact unit weighing approximately 300 g. Data are acquired and stored on the device for subsequent analysis.

The equipment can be set up quickly, is robust against motion and environmental noise, has high resolution, and is silent, which makes it suitable for neonatal use. The device is comfortable due to its soft optodes and rubber optode holders, which provide optimized flexibility for testing within neonatal settings. The device uses non-ionizing radiation and is non-invasive. The optodes will be placed on the neonatal scalp for a maximum of 45 min. During this short application duration, the likelihood of skin irritation is minimal. The Brite MKII is not licensed in Canada but has been authorized by Health Canada as an investigational device only for this research study (Figure 2a,b).

#### 2.7.2. Experimental Setup

We made the customized neonatal cap stitched in 3 sizes to snugly fit head circumferences of 33 cm, 35 cm, and 38 cm (Figure 3). The cap could be slightly stretched in all directions, allowing us to accommodate multiple head sizes and shapes without affecting the channels’ positions that correspond to the anatomical landmarks of the brain. We determined the optode positions based on the 10–20 international EEG system [35] for electrode placement. Each transmitter–receiver pair had a 2 cm distance between them as specified by the device (Figure 4). The cap had 20 emitters and 16 detectors, with 27 channels per side covering the frontal, parietal, and occipital lobes (Figure 4).

We will first set up the cap by placing the transmitter and receiver optodes into the optode holders based on the cap layout. We will start recording to take measurements during natural sleep. All infants will be breathing spontaneously in room air. No sedation will be given to the infant. Parents will be encouraged to be at the bedside, and the infant will be cuddled in a warm blanket before being placed in the cot. All recordings will be taken within two hours after feeding in a dimly lit room in the NICU. The total duration of the resting-state data acquisition will range from 30 to 45 min, depending on the infant’s cooperativeness.

#### 2.7.3. Data Acquisition

Before initiating data acquisition, the fNIRS recording device will be prepared by establishing a high-fidelity wireless connection with OxySoft software (OxySoft 4) and ensuring the device is ready for measurement. The recording device and the computer will be placed on a hospital cart and brought into the NICU. The cart is positioned adjacent to the crib, and the fNIRS device will be set beside the patient (Figure 2b). Next, the NIRS cap will be placed on the patient’s head and adjusted so that the central optodes align with the nasion and all optodes are positioned perpendicularly to the scalp with adequate contact. Once the cap is secured, an automatic signal calibration will be performed. A brief video of the patient wearing the cap will be recorded to co-register the recording channels with brain regions.

OxySoft is proprietary software specifically designed for facilitating the collection, storage, visualization, and analysis of fNIRS data. It supports the parallel recording of multiple devices, enabling synchronized data acquisition from multiple fNIRS units. In this study, OxySoft version 4.0 will record, manage, export, and monitor data in real time during measurements. OxySoft provides data acquisition (DAQ) status and DAQ values during signal calibration and data collection, conveying information such as device status, recording time, total recording duration, and the amount of light detected in each fNIRS recording channel. Channels whose light levels exceed or fall below acceptable thresholds are highlighted in red. Such out-of-range values may arise from head movements that displace the cap, environmental noise, or the presence of hair between the optode and the skin. If this occurs during signal calibration, the cap will be readjusted to optimize optode–scalp contact and calibration will be repeated until the light levels fall within the acceptable range. The research team will continuously monitor the DAQ values throughout the study.

#### 2.7.4. Neuroanatomy

The proposed study examines the interactions between different brain regions in neonates. Intrinsic brain activity at rest without a stimulus can provide information on resting-state functional connectivity, which yields Resting-State Networks (RSNs) [36]. Previously identified canonical RSNs in term neonates are located in primary motor and sensory cortices (sensorimotor), medial aspects of the occipital cortex (visual), the temporal lobe and inferior parietal cortex (auditory), and the association cortex (pro-default mode networks) [37,38]. The region of interest (ROI) will be defined manually on the age-specific templates by visual inspection using specific atlases of human brain development. From these ROI positions, seed time traces will be extracted for the three hemodynamic contrasts (HbO2, HbR and HbT). These time courses will then be correlated with every other voxel within the field of view to generate correlation (r) maps.

#### 2.7.5. Principle

DOT utilizes the absorption of infrared light by oxygenated and deoxygenated hemoglobin to measure changes in cortical blood flow, which indexes localized neural activity (Figure 5). This indirect measure of neural activity is similar to that used in fMRI, which has been widely used in adult and child neuroimaging studies based on neurovascular coupling. Neurovascular coupling refers to the process of the brain regulating blood flow in response to neural activity, which is a mechanism of ensuring there is enough oxygen to support higher-activity brain regions. Functional connectivity measures enable the determination of the relationship between neural demand and neurovascular coupling [39]. Therefore, measuring regional blood flow can provide information about the brain’s regulatory response to injury in the context of HIE. Near-infrared light is absorbed by oxygenated and deoxygenated hemoglobin in the cerebral vascular bed. A transmitter (source) emits light of a specific wavelength. The light passes through the cerebral vascular bed to be captured in a receiver (detector), which is also placed on the scalp’s surface. Standard light absorption properties of oxy- and deoxyhemoglobin are then used to calculate the amount of oxy- versus deoxyhemoglobin in the cerebral vascular bed. These values indirectly provide a measure of cerebral blood flow in different regions of the brain. Using the above principle, the raw fNIRS data collected are then processed through designated software to map the brain’s active and inactive regions in a stipulated time period. These assessments can be conducted at rest and in response to specific stimuli such as sound or light.

### 2.8. Objectives and Outcomes

#### Objectives and Corresponding Outcomes of the Feasibility Trial

Objective 1: To determine the feasibility of recruitment and retention strategies and data collection methods in the NICU. Feasibility will be assessed by reporting the following metrics: (i) The recruitment rate will be reported to assess the ability to recruit the target population within the desired timeframe, determined by the number of eligible participants who are enrolled in the study. We will maintain a recruitment tracking log. (ii) The follow-up rate will be used to assess the ability to retain participants and complete follow-up assessments, as determined by the percentage of enrolled patients who complete follow-up at 12 months. We will maintain a record of participant follow-up procedures and dropout rates. (iii) The rate of missing data will be determined by the percentage of completed data fields measured by reviewing collected data to identify any missing information, inconsistencies, or incomplete responses. *Criteria for Feasibility:* The main study will be feasible if the pilot study is successful based on the predefined criteria: (i) 80% of the eligible patients consent to the testing; (ii) 90% of the consented participants get DOT measurements taken before 7 days of life; (iii) for 80% of the consented participants, parent-reported developmental assessment is available at 6 and 12 months.

Objective 2: To assess the fidelity and safety of DOT measurements in neonates. (i) Safety will be assessed by reporting any adverse skin reactions to sensors or excessive agitation in the neonate, requiring early termination of the study. (ii) Data quality will be assessed by recording the degree of the signal-to-noise ratio. We will keep a record of the ability to properly place sensors with good contact and the ability to minimize motion artifacts and ambient light interference in each subject. (iii) Ability to complete adequate data acquisition within 45 min. *Criteria for Feasibility:* Time required for resting-state data acquisition per infant < 45 min.

Objective 3: To identify logistic challenges, unexpected adverse effects, or technical difficulties. Operational feasibility will be assessed based on the time and personnel required for each stage of the study, as well as the logistical challenges.

Objective 4: To gain stakeholder engagement, including that of parents and NICU nurses. The research team will also approach parents and bedside nurses at the end of the NICU stay with a brief survey to gather their perceptions about the study procedures. The survey consists of three questions: (1) Parents Section: Please let the study investigators know if you had any concerns during the DOT measurements of your child: (a) it takes a long time to measure, (b) the infant feels discomfort, (iii) others (free text), (iv) none. (2) Nursing Section: Please let the study investigators know if you had any concerns during the DOT measurements of your patient: (a) any concern from nursing staff about interfering with infant sleep, (b) interfering with the infant feeding schedule, (c) causing discomfort to the infant, (d) taking a long time, (e) the need for the nurse to be at the bedside, (f) others (free text), and (g) none.
Secondary objectives include the following:


Objective 5: To provide preliminary data on outcome measures to calculate the sample size of the adequately powered study. Clinical outcomes of the definitive trial will be measured to assess the reliability of data collection tools. Short-term neurological outcomes of neonates with HIE will be assessed using a brain MRI conducted within 7 days of life. The severity of brain injury on the MRI brain will be quantified using a detailed scoring system. The MRI score consists of 4 subscores, including grey matter (basal ganglia, thalamus, PLIC, brainstem, perirolandic cortex, and hippocampus), white matter/cortex (including optic radiation and corpus callosum), and cerebellum. Each category is weighted by its degree (0 = no injury; 1 = focal or <50%; 2 = extensive or >50%) and its location (1 = unilateral; 2 = bilateral). Additional scores are added if there is an intraventricular hemorrhage, subdural hemorrhage, or sinovenous thrombosis (score of 1 each if present). The final score represents a summation of all 4 subscores (maximum grey matter subscore 23, maximum white matter subscore 21, maximum cerebellum subscore 8, and maximum additional subscore 3), with 2 additional scores being added when H-MRS is performed in the basal ganglia and thalamus (score of 1 each if present: reduced N-acetyl aspartate (NAA) peak and increased lactate peak). This adds up to a maximum score of 57 [40].

Long-term neurological outcomes will be measured with a validated parent-reported questionnaire at 6 months and 12 months of age. We will use the Ages and Stages Questionnaire—3rd edition (ASQ-3) to assess five areas of childhood development—Communication, Gross Motor, Fine Motor, Problem-Solving, and Personal–Social—through parent/caregiver reports. Filling in the report takes 10–15 min, and it takes 2–3 min to score. The ASQ-3 is a widely used screening measure and has proven to be an effective developmental screening tool in cross-cultural populations [41]. The questionnaire includes 30 items, scored as yes, sometimes, or not yet, on questions about a child’s ability to perform a task [42]. The concurrent validity of the ASQ-3 with standardized testing of developmental (Bayley Scales of Infant Development) and intellectual (Stanford–Binet Intelligence Test–4th edition) skills has been demonstrated [43]. The ASQ-3 has a sensitivity of (0.70–0.90) and specificity of (0.76–0.91) in detecting developmental impairment, with retest reliability of (0.94–0.95) and interrater reliabilities between parents and professionals of (0.94–0.95) [44,45]. We will also use the Ages and Stages Questionnaire: socioemotional-2 (ASQ: SE-2) to perform a broadband social–emotional screening for study participants. Social–emotional domains include self-regulation, compliance, social communication, adaptive functioning, autonomy, affect, and interaction with people. ASQ: SE-2 test characteristics reflect an overall sensitivity of 81% across age intervals and an overall specificity of 84% across age intervals, with 89% test–retest reliability and 84% internal consistency [46]. Table 1 summarizes the study objectives, outcomes and corresponding data analysis. 

### 2.9. Participant Timeline

The planned duration of the study is 24 months. Details about the data collection timeline are provided in Table 2. The last 6 months will be dedicated to manuscript writing, preparation of final reports, and dissemination of knowledge. Collaborative meetings will be held monthly throughout the entire study period.

### 2.10. Sample Size and Recruitment

The National Center for Complementary and Integrative Health notes that sample size should be based on practical considerations, including participant flow, budgetary constraints, and the number of participants needed to reasonably evaluate feasibility goals [47]. Hence, for this work, a sample size of 25 was considered adequate to reach saturation and establish feasibility. If the feasibility criteria are met, the sample size for the larger trial will be determined based on the data analysis.

### 2.11. Clinical Data Collection

Relevant clinical data collected per study subject will include the following: (i) Maternal demographics: maternal age, pregnancy complications, multiparity, smoking, medication, alcohol, drug abuse, mental illness, maternal infections, and education level. (ii) Intrapartum variables: mode of delivery, duration of the second stage of labour, risk factor for sepsis, Apgar scores, details of resuscitation, inborn/outborn. The following details will be collected about the NICU course and complications: (i) hypothermia therapy: time of initiation, time to reach the target temperature; (ii) severity of encephalopathy: modified Sarnat Staging; (iii) severity of end-organ dysfunction: MODS scoring system; (iv) seizures, anti-epileptic drugs; (v) length of stay, No. of days on a ventilator; (vi) need for assisted feeding (NG/G-tube), home oxygen, tracheostomy; (vii) neurophysiology: background pattern of amplitude-integrated EEG at 6 h, 24 h, 48 h, and 72 h.

## 3. Data Processing

We will follow a standard data-processing pipeline to analyze the collected fNIRS data. First, the raw data will be exported from OxySoft in the Shared Near-Infrared File (SNIRF) format, which retains optode positions and timestamps. Subsequently, the open-source NIRS data analysis software HOMER3, version 1.80.2 [48], will be employed to perform a series of preprocessing steps, including converting raw intensity data to hemoglobin concentration changes based on the modified Beer–Lambert Law [48]. The data preprocessing pipeline consists of the following steps:(1)Converting raw light intensity data to optical density change data;(2)Excluding channels with out-of-range signals or large standard deviations;(3)Correcting motion artifacts using Temporal Derivative Distribution Repair (TDDR), a regression-based approach that addresses motion-induced spikes and baseline drift [49];(4)Band-pass filtering (0.01–0.05 Hz) to eliminate low-frequency noise (e.g., data drift) and high-frequency physiological noise (e.g., cardiac signals);(5)Converting the preprocessed optical density data to hemoglobin concentration change data using a Differential Path Length Factor (DPF) of 6 for both oxy- and deoxyhemoglobin.

After preprocessing, resting-state functional connectivity and network analyses will be conducted using NIRS_KIT [50] and the Graph Theoretical Network Analysis Toolbox (GRETNA) [51], both of which are MATLAB-based open-sourced toolboxes. Pearson correlation coefficients will be calculated between all pairs of channels and among channels within specific ROIs. GRETNA will then analyze these correlation coefficients to characterize the topology of both whole-brain and ROI-specific networks. To investigate disruptions in specific neural networks, we will focus on the functional connectivity of the default mode, sensorimotor, auditory, and visual networks. These networks will be identified by co-registering the fNIRS recording channels to the standard MNI space using STORM-Net [52], a Python-based photogrammetry application (Version 3.13). We will then calculate global network metrics including **assortativity** (the tendency of nodes to connect with other nodes of similar degree), **hierarchy** (the degree of hierarchical organization), **small-worldness** (the ratio between local clustering and global integration), **global efficiency** (the efficiency of parallel information transfer), **local efficiency** (the network’s fault tolerance), and **synchronization** (the degree of correlated activity among nodes) [53]. These metrics provide insights into communication efficiency, information processing speed, network resilience, and overall connectivity. We hypothesize that hypoxic–ischemic injury will alter these network metrics, specifically manifesting as weaker interhemispheric connectivity and stronger intrahemispheric connectivity, which may represent compensatory adaptations during recovery.

## 4. Potential Challenges and Limitations

DOT does not provide information on the subcortical structures; only connectivity between different regions of the cerebral grey matter is measured. Hypoxic–ischemic brain injury may affect the microstructure of the white matter tracts and deep grey matter. Structural connectivity between brain regions is better assessed by diffusion tensor MR imaging. Our results will not provide information about the deep grey matter structures such as the basal ganglia and thalamus. Future multi-modal approaches, such as DOT combined with EEG and MRI, may overcome the challenge of individual modality limitations. The limitation of most pediatric optical systems is the inability to place a full array of optical probes just as in adults. In adults, high-density DOT (64 channels on each side) achieves improved spatial resolution and better separation of cerebral signals from superficial confounds. The translation of high-density DOT techniques to neonates is a challenging task. There is a need for cap ergonomics to maximize comfort while enhancing the coupling between the optical probes and the infant’s scalp. Since this is a feasibility study, we initially planned to start with 27 channels. If acceptable signal quality is not achieved, we may try (a) increasing the number of optodes used or (b) using subject-specific structural MRIs to validate the individualized DOT reconstructions relative to subject-matched MRI. Combining atlas transformation techniques with age-matched neonatal templates could provide realistic head models using a nonlinear warping procedure.

The small sample size may provide a false sense of any association between functional connectivity and prognostication in neonates with HIE. Given the nature of HIE and the sensitivity of such discussions with parents, a challenge that may arise during this study is obtaining informed consent from parents and substitute decision makers during a time when the prognosis of their baby is uncertain. Many parents may not want to discuss any intervention that will not benefit their baby. As participants are recruited, it will be determined whether our approach to the consent process needs to be altered to have respectful conversations with parents.

## 5. Discussion

This study examines the potential for more accurate neurocognitive prediction methods in the early neonatal period to inform clinical decision making and early intervention. The opportunity for DOT to assess functional connections between brain regions and quantify region-specific brain activity at the bedside could open up many doors for functional prognostication in HIE. However, the feasibility of such a study needs to be assessed before conducting a clinical trial to evaluate this new technology. Additionally, it is essential to understand the relationship between resting-state functional connectivity measures, assessed via DOT, and neurological outcomes. Given the prevalence of HIE and its sequelae, determining if DOT can be used to prognosticate neurological outcomes would have significant implications for future neonatal care. Another future direction for this study would be to add EEG electrodes to the DOT caps and record fNIRS and EEG simultaneously. Quantitative EEG abnormalities in this cohort of neonates are being explored for prognostic utility [54]. Since EEG allows for source localization, including during sleep, combining these two methodologies could facilitate further localization of cerebral oxygenation during sleep–wake cycles [55].

## 6. Conclusions

Perinatal brain injury is a complex condition in neonates associated with significant future implications for neurological function in later childhood. There is a need for more effective prognostic tools that can help clinicians identify changes in brain function in the early stages of life, enabling the development of tailored early intervention strategies to improve future outcomes. This study investigates the potential clinical use of a non-invasive bedside tool that can provide additional information about the injured neonatal brain, complementing the information from structural neuroimages.

## Figures and Tables

**Figure 1 neurosci-06-00060-f001:**
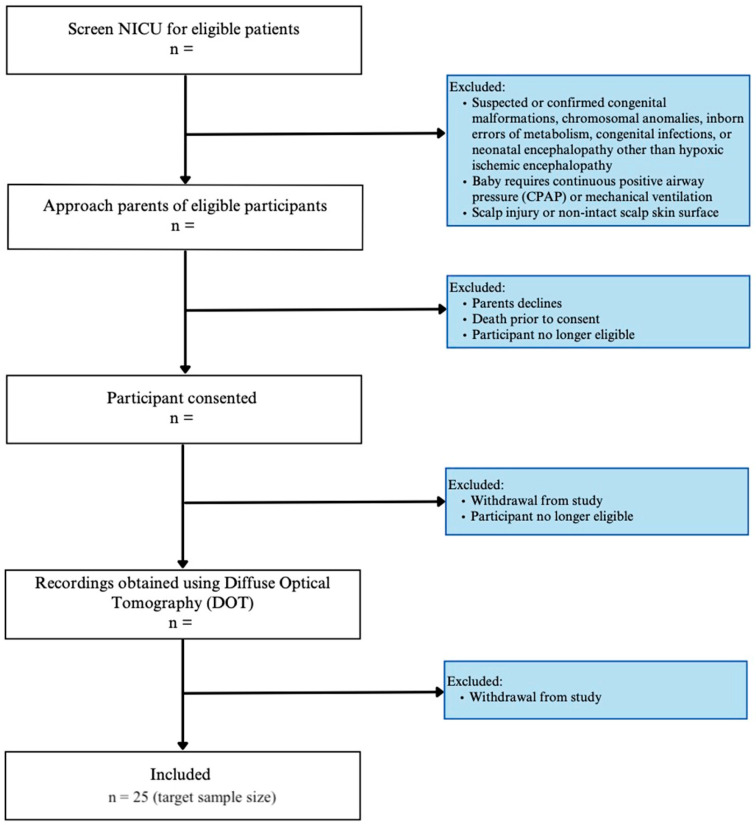
Study enrollment flow diagram.

**Figure 2 neurosci-06-00060-f002:**
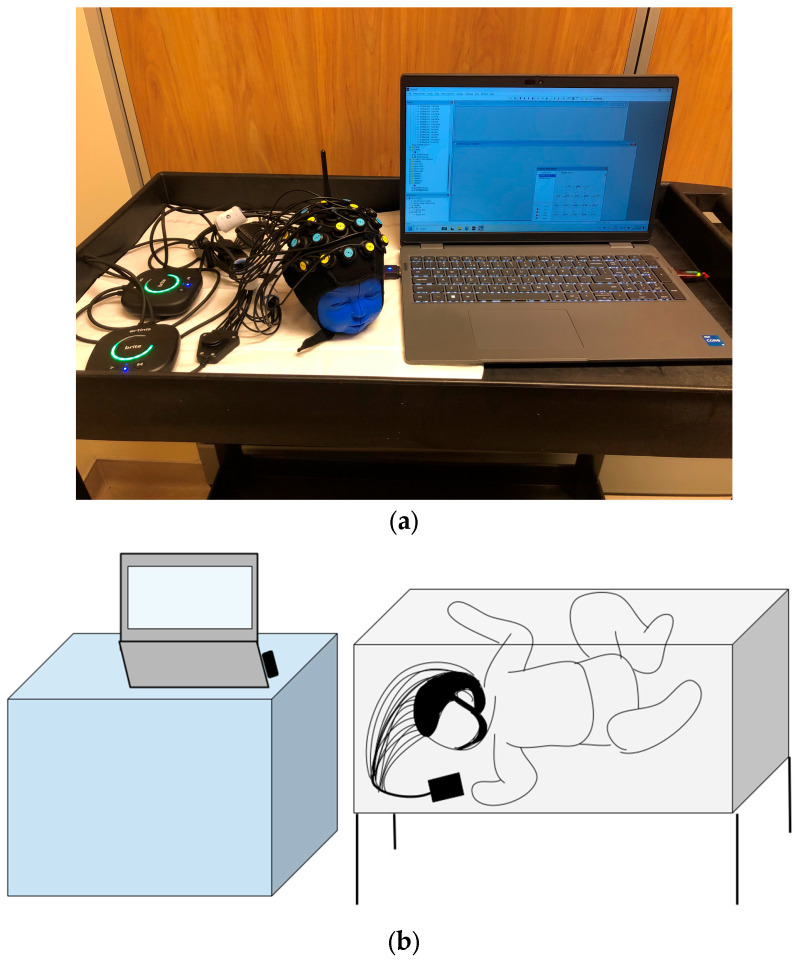
(**a**) The experimental setup showing the neonatal cap with sensors attached placed on a mannequin, along with two fNIRS devices and the Laptop that is used for remote monitoring. (**b**) An illustration of the device being used in the NICU bedside. A Bluetooth-enabled system with a low footprint can be comfortably brought to the NICU bedside.

**Figure 3 neurosci-06-00060-f003:**
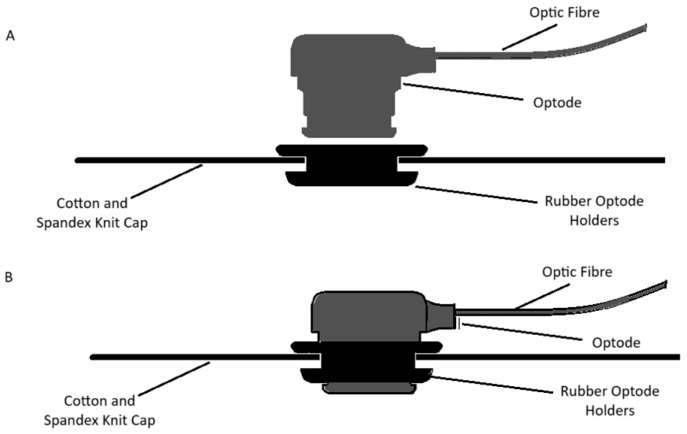
Customized neonatal caps with and without the optode. The fabric used for the cap is 5% spandex and 95% cotton knit. The soft rubber optodes ensure the cap is gentle on the neonatal scalp (**A**). The optodes are slid into the holders until they are flush with the holders, which ensures good contact with the neonatal scalp (**B**).

**Figure 4 neurosci-06-00060-f004:**
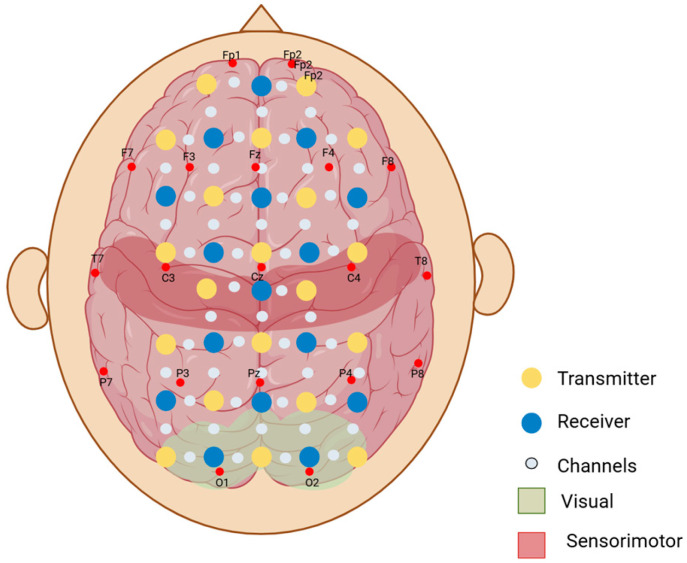
Representation of the two-dimensional layout of the optodes on the scalp. Yellow optodes represent the transmitters (optodes that emit the near-infrared light), and the blue optodes represent the receivers (optodes that detect the light scattered through the cortex and back). Small white-grey dots represent the channels between each transmitter and receiver pair (optical paths where the fluctuations in near-infrared light are measured). F, T, P, C, and O refer to the anatomical landmarks (frontal, temporal, parietal, central, and occipital, respectively) according to the 10–20 electrode placement system.

**Figure 5 neurosci-06-00060-f005:**
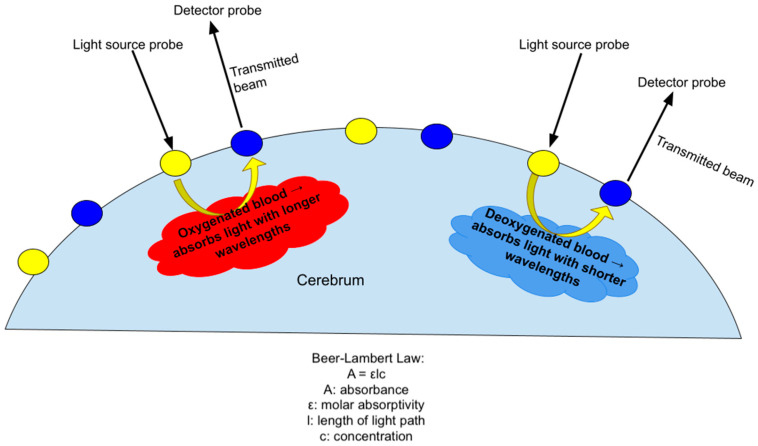
Demonstration of the principle of DOT, which uses differing absorption of infrared light by deoxygenated and oxygenated hemoglobin to measure changes in cortical blood flow in accordance with the Beer–Lambert Law.

**Table 1 neurosci-06-00060-t001:** Summary of the study objectives, outcomes, and analysis plans.

Objectives	Outcomes	Criteria for Success	Method of Analysis
Primary Aims			
Assess the feasibility of recruitment and retention strategies and data collection methods in the NICU.	Recruitment and retention rates, missing data.	(i) 80% of the eligible patients consent to the testing; (ii) 90% of the consented participants get DOT measurements taken before 7 days of life; (iii) for 80% of the consented participants, parent-reported developmental assessment is available at 6 and 12 months; (iv) Time required for resting-state data acquisition per infant < 45 min.	Descriptive statistics (mean and SD) for continuous variables. Proportions for dichotomous variables.
To assess the fidelity and safety of DOT measurements in neonates.	Adverse reactions, adequate measurements, and time for data acquisition.	(i) Reporting any adverse skin reactions to sensors or excessive agitation in the neonate requiring early termination of the study. (ii) Data quality will be assessed by recording the degree of the signal-to-noise ratio.	Qualitative analysis.
To identify logistic challenges, unexpected adverse effects, or technical difficulties.	Adverse events or operational difficulties.	An adequate amount of time and number of personnel required for each stage of the study and staff workload. Lack of logistic challenges.	Qualitative reporting.
To gain stakeholder engagement, including from parents and NICU nurses.	Engagement level.	Positive perceptions about the study procedures from parents and nurses.	Qualitative survey analysis.
**Secondary Aims**			
**Objectives**	**Outcome**	**Hypothesis**	**Analysis**
To explore the differences in brain metrics between male–female, mild–moderate–severe encephalopathy groups.	Brain metrics.	Significant difference in regional brain metrics.	Student *t*-test.
To explore the correlation of severity of brain injury score with brain metrics.	MRI injury score.	We can determine the required sample size for a definitive trial.	Regression analysis.
Correlation of brain metrics with neurological outcome.	ASQ scores at 6 months and 12 months.	We can determine the required sample size for a definitive trial.	Regression analysis.

**Table 2 neurosci-06-00060-t002:** Participant timeline.

	Study Period								
	Enrollment	Allocation	Post-Allocation						Close Out
Timepoint		Day 0	1–3 days	4–7 days	4–15 days	NICU discharge	6 months	12 months	
Screening for eligibility	X								
Approaching parents for informed consent	X								
Study ID created	X								
Allocation	X								
Clinical data collection	X	X	X	X	X	X			
Therapeutic hypothermia			X						
EEG monitoring			X						
MRI brain				X					
DOT measurement					X				
ASQ-3							X	X	
ASQ: SE-2							X	X	

Abbreviations: ASQ-3, Ages and Stages Questionnaire-3rd version; ASQ: SE-2 Ages and Stages Questionnaire: Socio-emotional-2nd version; MRI, magnetic resonance imaging; DOT, diffuse optical tomography; EEG, electroencephalogram.

## Data Availability

Not applicable.

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
