# Peer review of "Derivation of Novel Imaging Biomarkers of Neonatal Brain Injury Using Bedside Diffuse Optical Tomography: Protocol for a Prospective Feasibility Study"

_neurosci, 2025, doi:10.3390/neurosci6030060_

Round 1
Reviewer 1 Report
Comments and Suggestions for Authors
This is an excellent protocol and I would support its publication with the following revision of the present manuscript:
1) In the study setting section I would kindly ask to specify for non-NorthAmerican readers that Level III centers equal to Major Trauma Centre in the UK and Europe. Hence aligning with the requirements listed by Dasic et al. (namely: availability of all surgical and medical subspecialties for adult and paediatric pathologies, availability of general and neuro-intensive care for adult and paediatric patients, availability of neurointerventional radiology, direct hub and spoke system with surrounding district general hospitals and possibility to transfer acute patients via helipad)
Ref: Dasic, D.; Morgan, L.; Panezai, A.; Syrmos, N.; Ligarotti, G.K.I.; Zaed, I.; Chibbaro, S.; Khan, T.; Prisco, L.; Ganau, M. (2022) A scoping review on the challenges, improvement programs, and relevant output metrics for neurotrauma services in major trauma centers. Surg Neurol Int. 13, 171. doi: 10.25259/SNI_203_2022.
2) I understand that calculating a sample size for an exploratory study is challenging but I would still believe that the quality of the present manuscript would be increased by adding a paragraph with your considerations. I could suggest stating that: "the National Center for Complementary and Integrative Health (NCCIH) notes that sample size should be based on practical considerations including participant flow, budgetary constraints, and the number of participants needed to reasonably evaluate feasibility goals. Hence, for qualitative and quantitative works, such as this one, sample sizes of 30 cases or less may be adequate to reach saturation and establish feasibility."
Ref: Teresi, J.A.; Yu, X.; Stewart, A.L.; Hays, R.D. (2022) Guidelines for Designing and Evaluating Feasibility Pilot Studies. Med Care. 60(1), 95-103. doi: 10.1097/MLR.0000000000001664
3) I approve the focus on imaging biomarkers but would be grateful if you could reference the works currently conducted to establish other biosignatures of acquired brain injury following neonatal hypoxic-ischemic events, including the Dammiss EEG score which would complement well your methods section or if no changes to current protocol are possible this score should be mentioned in passing in your discussion.
Ref: Ferrari, F.; Bondi, C.; Lugli, L.; Bedetti, L.; Guidotti, I.; Banchelli, F.; Lucaccioni, L.; Berardi, A. (2025) The Dammiss EEG Score: A New System to Quantify EEG Abnormalities and Predict the Outcome in Asphyxiated Newborns. J Clin Med. 14(6):1920. doi: 10.3390/jcm14061920.
I hope that those suggestions will facilitate the revision process and help the authors in preparing a revised submission at their earliest convenience.
Author Response
Comment 1: This is an excellent protocol and I would support its publication with the following revision of the present manuscript: 1) In the study setting section I would kindly ask to specify for non-NorthAmerican readers that Level III centers equal to Major Trauma Centre in the UK and Europe. Hence aligning with the requirements listed by Dasic et al. (namely: availability of all surgical and medical subspecialties for adult and paediatric pathologies, availability of general and neuro-intensive care for adult and paediatric patients, availability of neurointerventional radiology, direct hub and spoke system with surrounding district general hospitals and possibility to transfer acute patients via helipad). Ref: Dasic, D.; Morgan, L.; Panezai, A.; Syrmos, N.; Ligarotti, G.K.I.; Zaed, I.; Chibbaro, S.; Khan, T.; Prisco, L.; Ganau, M. (2022) A scoping review on the challenges, improvement programs, and relevant output metrics for neurotrauma services in major trauma centers. Surg Neurol Int. 13, 171. doi: 10.25259/SNI_203_2022.
Response 1: Thank you for pointing this out. We agree with this comment. Therefore, we have added the following on Page 3 under Study Setting: "The province of Ontario, Canada perinatal services are organized into three levels namely, (i) Level 1: low neonatal risk, postnatal care of healthy newborn and resuscitation/stabilization of ill infants before transfer; (ii) Level 2: neonatal care for gestational age > 32 weeks and birth weight > 1500g [exception: Level 2c >30 weeks and > 1200g] and (iii) Level 3: neonatal intensive care for any gestational age and birth weight, availability of subspecialty consultants and surgical capability. Since therapeutic hypothermia resources are available solely in Level 3 hospitals in Ontario, Canada, neonates born across Level 1 and 2 hospitals are transferred to Level 3 with a specialized neonatal transport team [www.pcmch.on.ca]. McMaster Children’s Hospital McMaster Children’s Hospital is a Level III unit attached to a high-risk maternal and fetal medicine unit."
Comment 2: I understand that calculating a sample size for an exploratory study is challenging but I would still believe that the quality of the present manuscript would be increased by adding a paragraph with your considerations. I could suggest stating that: "the National Center for Complementary and Integrative Health (NCCIH) notes that sample size should be based on practical considerations including participant flow, budgetary constraints, and the number of participants needed to reasonably evaluate feasibility goals. Hence, for qualitative and quantitative works, such as this one, sample sizes of 30 cases or less may be adequate to reach saturation and establish feasibility." Ref: Teresi, J.A.; Yu, X.; Stewart, A.L.; Hays, R.D. (2022) Guidelines for Designing and Evaluating Feasibility Pilot Studies. Med Care. 60(1), 95-103. doi: 10.1097/MLR.0000000000001664
Response 2: Thank you for pointing this out. We agree with this comment. Therefore, we have added the following on Page 12, under 2.10 (Sample Size and Recruitment): "The National Center for Complementary and Integrative Health (NCCIH) notes that sample size should be based on practical considerations including participant flow, budgetary constraints, and the number of participants needed to reasonably evaluate feasibility goals. Hence, for this work sample sizes of 25 was considered adequate to reach saturation and establish feasibility.[47].
Comment 3: "I approve the focus on imaging biomarkers but would be grateful if you could reference the works currently conducted to establish other biosignatures of acquired brain injury following neonatal hypoxic-ischemic events, including the Dammiss EEG score which would complement well your methods section or if no changes to current protocol are possible this score should be mentioned in passing in your discussion."
Response 3: Thank you for pointing this out. We agree with this comment. Therefore, we have added the following on Page 15, Line 525-526: "Quantitative EEG abnormalities in this cohort of neonates is being explored for prognostic utility.[54]"
Reviewer 2 Report
Comments and Suggestions for Authors
Management of neonatal hypoxic-ischaemic brain injury/lesion falls under the purview of major and high-specialized trauma centre in Europe and this should be clearly stated/stressed in the study setting paragraph along with a reference for the requirements to classify hospitals as MTC or Level III (e.g. DOI: 10.25259/SNI_203_2022). Otherwise such prospective study is well written and designed, methods sound well and the topic quite original. I believe that it could be reconsidered after minor revision.
Author Response
Comment 1: "Management of neonatal hypoxic-ischaemic brain injury/lesion falls under the purview of major and high-specialized trauma centre in Europe and this should be clearly stated/stressed in the study setting paragraph along with a reference for the requirements to classify hospitals as MTC or Level III (e.g. DOI: 10.25259/SNI_203_2022). Otherwise such prospective study is well written and designed, methods sound well and the topic quite original. I believe that it could be reconsidered after minor revision."
Response 1: Thank you for pointing this out. We agree with this comment. Therefore, we have added the following on Page 3, under Study Setting: Page 3 under Study Setting: "The province of Ontario, Canada perinatal services are organized into three levels namely, (i) Level 1: low neonatal risk, postnatal care of healthy newborn and resuscitation/stabilization of ill infants before transfer; (ii) Level 2: neonatal care for gestational age > 32 weeks and birth weight > 1500g [exception: Level 2c >30 weeks and > 1200g] and (iii) Level 3: neonatal intensive care for any gestational age and birth weight, availability of subspecialty consultants and surgical capability. Since therapeutic hypothermia resources are available solely in Level 3 hospitals in Ontario, Canada, neonates born across Level 1 and 2 hospitals are transferred to Level 3 with a specialized neonatal transport team [www.pcmch.on.ca]. McMaster Children’s Hospital McMaster Children’s Hospital is a Level III unit attached to a high-risk maternal and fetal medicine unit."
Reviewer 3 Report
Comments and Suggestions for Authors
This study protocol outlines a well-conceived and methodologically sound pilot study designed to assess the feasibility of using diffuse optical tomography (DOT) as a bedside tool for evaluating functional brain connectivity in neonates with hypoxic-ischemic encephalopathy (HIE). The manuscript is clearly written, thoroughly referenced, and addresses an important gap in neonatal neuroprognostication, offering an innovative, non-invasive alternative to conventional neuroimaging techniques. The proposed use of DOT in the early neonatal period for functional prognostication is both innovative and clinically relevant. The approach is justified by current limitations of MRI and EEG in predicting long-term outcomes in HIE. The manuscript is well-organized and clearly written.
My minor suggestions:
- The limitations section appropriately notes that DOT cannot assess subcortical structures. A brief discussion on how future multi-modal approaches (e.g., DOT + EEG or DOT + MRI fusion) might overcome this could enhance the translational relevance.
- A few minor typographical inconsistencies were noted (e.g., spacing around percentages, use of “°C” formatting). A careful proofread will resolve these.
Author Response
Comment 1: "The limitations section appropriately notes that DOT cannot assess subcortical structures. A brief discussion on how future multi-modal approaches (e.g., DOT + EEG or DOT + MRI fusion) might overcome this could enhance the translational relevance."
Response 1: Thank you for pointing this out. We agree with this comment. Therefore, we have added the following on Page 14, line 490-491-520: "Future multi-modal approaches, such as DOT combined with EEG and MRI fusion, may overcome the challenge of individual modality limitations."
Comment 2: "A few minor typographical inconsistencies were noted (e.g., spacing around percentages, use of “°C” formatting). A careful proofread will resolve these."
Thank you for pointing this out. We agree with this comment. We have made the edits, found on multiple pages of the document, including: page 2, line 98 (space removal); page 4, line 151-152 (fixed degrees Celsius formatting); page 11, line 381 (adjusted r2 superscript), line 390, 393 and 402 (removed spaces); page 13, line 466 (removed capital letter in "assortativity". The new manuscript has highlighted each of the new edits.
Round 2
Reviewer 1 Report
Comments and Suggestions for Authors
The authors revised well, although they did not include Dasic matrix of unmet needs in their revised version. I think this is a minor issue but worth mentioning. I will leave to their discretion the decision on whether to add it or not, I will grant my approval for publication as of now so there is no need for further revision.
Author Response
Comment 1: "The authors revised well, although they did not include Dasic matrix of unmet needs in their revised version. I think this is a minor issue but worth mentioning. I will leave to their discretion the decision on whether to add it or not, I will grant my approval for publication as of now so there is no need for further revision. "
Response 1: Thank you for pointing this out. The Dasic matrix of unmet needs is generally for implementation research projects; hence, it does not apply directly to our study protocol, so we have decided not to add it in our manuscript.